

# The contribution of air temperature and ozone to mortality rates during hot weather episodes in eight German cities during the years 2000 and 2017

Alexander Krug[1,2], Daniel Fenner[1], Hans-Guido Mücke[2], and Dieter Scherer[1]

[1]Technische Universität Berlin, Institute of Ecology, Chair of Climatology, 12165 Berlin, Germany
[2]German Environment Agency, Section II 1.5 Environmental Medicine and Health Effects Assessment, 14195 Berlin, Germany

**Correspondence:** Alexander Krug (a.krug@tu-berlin.de)

**Abstract.** Hot weather episodes are globally associated with increased mortality. Elevated ozone concentrations occurring simultaneously contribute to mortality during these episodes, yet to what extent both stressors are linked to increased mortality rates varies from region to region.

This study analyzes time series of observational data of air temperature and ozone concentrations for eight German cities during the years 2000 and 2017. By using an event-based risk approach, various air temperature thresholds were explored for each city to detect hot weather episodes which are statistically associated with increased mortality. Multiple linear regressions were calculated to investigate the relative contribution of air temperature and ozone concentrations to mortality rates during these episodes, including their interaction. Results were compared for their similarities and differences among the investigated cities.

In all investigated cities hot weather episodes, linked to increased mortality rates, were detected. Results of the multiple linear regression further point towards air temperature as the major stressor explaining mortality rates during these episodes by up to 60 %, and ozone concentrations by up to 20 %. The strength of this association both for air temperature and ozone varies across the investigated cities. An interactive influence was found between both stressors, underlining their close relationship. For some cities, this interactive relationship explained more of the observed variance in mortality rates than each individual stressor alone.

We could show that during hot weather episodes, not only air temperature affects urban populations. Concurrently high ozone concentrations also play an important role for public health in German cities.

## 1 Introduction

Hot weather episodes (HWE) cause more human fatalities in Europe than any other natural hazard (EEA, 2019). HWE are typically characterized by elevated air temperature and can last for several days or weeks, depending on respective threshold values that are used to identify such days. Numerous investigations found excessive mortality rates during days of elevated air temperature (Curriero et al., 2002; Anderson and Bell, 2009; Gasparrini and Armstrong, 2011; Gasparrini et al., 2015).



Increases in morbidity rates, hospital admissions and emergency calls are also associated with elevated air temperatures (Bassil et al., 2009; Karlsson and Ziebarth, 2018).

25   In addition, HWE are linked to increased tropospheric ozone concentrations (Shen et al., 2016; Schnell and Prather, 2017; Phalitnonkiat et al., 2018). Zhang et al. (2017) and Schnell and Prather (2017), e.g., found for North America that the probability is up to 50 % that both air temperature and ozone concentrations reach their 95th percentile simultaneously. Ozone as a secondary air pollutant is formed by oxidation of volatile organic compounds. Increased air temperature and high solar radiation intensify this formation (Camalier et al., 2007; Varotsos et al., 2019). Correlations between both environmental stressors are mostly described as linear (Steiner et al., 2010). A variety of geographic and meteorological factors may influence this relationship, such as the presence of precursors, local-specific wind patterns or the humidity content of the lower atmosphere (Steiner et al., 2010). At the upper end of the respective air temperature and ozone concentration distributions the direct linkage between the two stressors is discussed to be even more complex (Steiner et al., 2010; Shen et al., 2016). Despite this linkage, elevated ozone concentrations alone have also been associated with adverse health effects (Bell, 2004; Hůnová et al., 2013; Bae et al., 2015; Díaz et al., 2018; Vicedo-Cabrera et al., 2020). The close linkage of both environmental stressors makes it necessary to account for their confounding influence on each other, in order to investigate distinctive health effects of each of these two stressors. But beyond the consideration of both environmental stressors as separated elements, their co-occurrence may lead to even higher rates of excess mortality (Burkart et al., 2013; Vanos et al., 2015; Scortichini et al., 2018; Krug et al., 2019). Some studies also indicate an interactive effect, which is larger than the sum of their individual effects (Cheng and Kan, 2012; Burkart et al., 2013; Analitis et al., 2018).

Studies which investigate regional differences in the relation between HWE and ozone concentrations revealed differences in the air-temperature-ozone relationship (e.g., Shen et al., 2016; Schnell and Prather, 2017; Phalitnonkiat et al., 2018) and in terms of their individual and combined effects on mortality (Filleul et al., 2006; Burkart et al., 2013; Analitis et al., 2014; Breitner et al., 2014; Tong et al., 2015; Analitis et al., 2018; Scortichini et al., 2018). Some studies report a North-South gradient in the air-temperature-mortality relationship, indicating that populations of northern regions are more sensitive to heat compared to southern regions, which are more affected by cold (e.g., Burkart et al., 2013; Scortichini et al., 2018). However, the influencing effect of elevated ozone concentrations is shown to be more differentiated. While some studies report a greater influence of elevated ozone concentrations for more heat-affected regions (Anderson and Bell, 2011; Scortichini et al., 2018), other studies discuss that regional differences are a result of location-specific physiological, behavioral, and social-economic characteristics, as well as the specific level of exposition across various cities (Anderson and Bell, 2009, 2011; Burkart et al., 2013; Breitner et al., 2014). For Germany, most studies investigated the effect of air temperature during HWE on mortality for different regions (Gabriel and Endlicher, 2011; Scherer et al., 2013; Muthers et al., 2017; an der Heiden et al., 2019). Breitner et al. (2014) investigated short-term effects of air temperature on mortality and modifications by ozone in three cities in southern Germany. But, to our knowledge, a national multi-city study exploring the impacts of HWE on mortality across different German cities has not been carried out so far. In addition, how ozone concentrations contribute to mortality rates during HWE are inconclusive for different cities in Germany and world-wide, as described above.





A prior study for Berlin, Germany (Krug et al., 2019), identified HWE and episodes of elevated ozone concentrations with a risk-based approach for the period 2000 to 2014. Whereas ozone concentrations alone only showed a weak relationship to mortality rates, the co-occurrence with elevated air temperatures amplified mortality rates in Berlin. On the basis of these results, main focus of this study lies on the identification of HWE in multiple cities in Germany and to investigate how air temperature and ozone concentrations contribute to mortality rates during these. Furthermore, the analysis period is extended up until 2017.

Main goals of this study are (a) to identify HWE that show statistical relations to mortality rates for eight of the largest German cities and (b) to compare these cities in terms of their location-specific relation of air temperature and ozone concentrations onto mortality rates. This study is structured by the following research questions:

1. Do other German cities, likewise Berlin, show a significant relationship between HWE and their specific mortality rates?

2. How does this relationship differ in terms of city-specific threshold values and the relative contribution of air temperature and ozone concentrations during HWE to the overall explained variance of the mortality rate?

## 2 Data and methods

### 2.1 Data

The period analyzed in this study are 18 years from 2000 to 2017. Eight cities are investigated (in the order of their population): Berlin, Hamburg, Munich, Cologne, Frankfurt (Main), Stuttgart, Leipzig and Hanover (Fig. 1, Table 1). While the first six are the six most populous cities in Germany, the latter two were included in this study to ensure spatially relatively homogeneous distribution of the investigated cities in Germany. The analyzed cities comprise 10.3 million inhabitants at the end of 2017, which were 12.5 % of the entire German population at this time (DESTATIS, 2019). The smallest city in terms of population (Hanover) has > 500 000 inhabitants, while the largest (Berlin) has > 3.5 million (Table 1).

### 2.1.1 Air temperature

Air temperature data at daily resolution was obtained from the German Weather Service (DWD, 2019). The selection of measurement sites was based on the availability of data covering the entire analysis period. For cities with more than one measurement site the site closest to the city center and to the co-located ozone measurement site was selected. An overview of the selected measurement sites including their meta data is given in Table A1 and Fig. A1 in the appendix. We use daily average air temperature (TA) from each station. Previous studies found this to be a suitable predictor for the air-temperature-mortality relationship and a suitable indicator for the city's diurnal thermal conditions, compared to maximum or minimum air temperature (Hajat et al., 2006; Anderson and Bell, 2009; Vaneckova et al., 2011; Yu et al., 2010; Scherer et al., 2013; Chen et al., 2015).

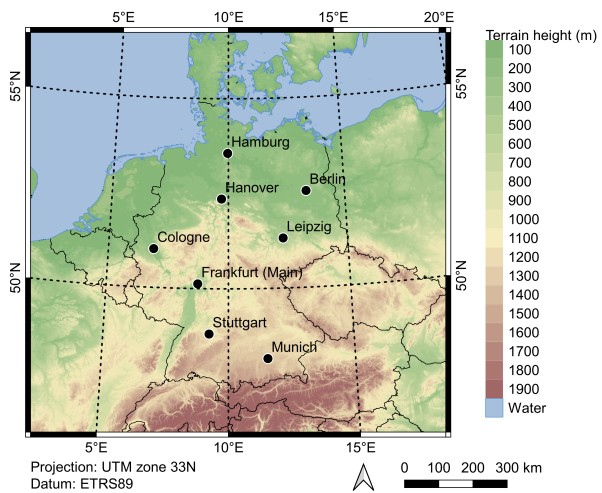

**Figure 1.** Location of investigated cites. Topographic map is based on GTOPO30 data retrieved from European Environment Agency (EEA, 2016)

### 2.1.2 Ozone concentrations

Data of hourly ozone concentrations were obtained from the German Environment Agency (UBA). These data stem from the air quality monitoring networks of the German federal states. To select one ozone monitoring station per city same criteria as for the selection of TA measurement sites were applied (closest to city center and to TA site). Only "urban background" stations were selected, as the ozone concentrations should be "representative of the exposure of the general urban population" (EU, 2008). The daily maximum eight-hour moving average (MDA8) was calculated from hourly values for all selected sites, which is the widely used metric for ozone monitoring for human health purposes (WHO, 2006; EU, 2008).

### 2.1.3 Population data

Time series of annual population counts were obtained for each city from the German Federal Bureau of Statistics (DESTATIS, 2019). The German census of 2011 revealed an error between 1 % and 5 % to the previously available annually updated version of the population time series for the selected cities, based on the prior census in 1990. Therefore, population time series were corrected based on the assumption that the error (a) increases over time and (b) correlates with the strength of the annual migration of each city. An error term was calculated for each year of the census period from 1990 to 2010 as the annual proportion of the total error (derived from the difference in the two census data in 2011). Each error term was further weighted by the proportion of the annual migration size from total migration size during the census period. This weighted error term was then subtracted from the annual population size. Years 2012 to 2017 were likewise corrected based on afore-mentioned assumptions. Annual time series were then linearly interpolated to daily values for each city.





### 2.1.4 Mortality data

Daily values of deaths for each city were provided by the German Federal Bureau of Statistics (DESTATIS, 2019). We in-
tentionally consider all-cause and all-age total death counts of the whole city in this study, as the main goal is to explore the
process which could have an effect (e.g. mortality) as a city-wide variable without any pre-assumption of disease-specific and
heat-related health effects. For that reason, we do not want to exclude any death counts from the analysis that might be related
to TA or MDA8. Mortality rates were calculated by dividing daily death counts by daily interpolated population counts.

Each time series of TA, MDA8, population and mortality rate were tested for a long-term annual trend. Whereas for TA
and MDA8 no significant long-term trend could be detected over the analysis period, mortality rates in all cities showed a
significant ($p < 0.05$, double-sided t test) negative annual trend. This trend was corrected for to avoid any misinterpretation of
the variance in the time series.

## 2.2 Methods

The methodological approach used in this study follows the concept of risk evaluation by the Intergovernmental Panel on Cli-
mate Change (IPCC) (IPCC, 2012). This concept was adopted for an explorative event-based risk analysis, which is explained
in detail by Scherer et al. (2013) and used in previous works to deduce two risk-based definitions of heat waves (Fenner et
al., 2019) or to quantify heat-related risks and hazards (Jänicke et al., 2018). This approach was also used to analyze the
co-occurrence of HWE and episodes of elevated ozone concentrations in Berlin (Krug et al., 2019). The main advantage of
this approach is that it explores time series without any pre-assumptions concerning threshold value, length or existing relation
between potentially hazardous episodes (here described with TA) and an effect variable (here the mortality rate). In order to
identify HWE with a significant relation to mortality rates, the approach as described in (Krug et al., 2019) was applied. In this
prior study, time series of TA and MDA8 were explored separately and the episodes, described as "events", were afterwards
classified as temporally separated or co-occurring events of elevated TA and MDA8. Deviating from that approach, only HWE
as characterized by elevated TA and identified by various threshold values are analyzed in this study. MDA8 is treated as an
additional stressor during HWE and analyzed as described in Sect. 2.2.2.

### 2.2.1 Detection of HWE

Firstly, time series of TA for each city were searched for HWE as the occurrence of at least three consecutive days exceeding
a certain TA threshold value ($TA_{Thres}$). $TA_{Thres}$ was iteratively increased in 0.5 K steps within the range 10 °C to 30 °C.
Secondly, at each $TA_{Thres}$ TA magnitude ($TA_{Mag}$) was calculated for each HWE as the accumulated sum of the difference of
daily TA and respective $TA_{Thres}$ over the whole length of the HWE (sum of degree days above $TA_{Thres}$). Thirdly, univariate
linear regressions were calculated between $TA_{Mag}$ as predictor variable (logarithmized) and mean mortality rates during the
HWE plus a maximum number of lag days (to account for possible lag effects in mortality rates after HWE) as the dependent
variable over the whole study period. Regression models thus consist of a unique combination of $TA_{Thres}$ and maximum lag
days. Models for each $TA_{Thres}$ were tested for a lag effect of maximum 0 to 7 days. Afterwards, the lag effect was fixed to four


days, which was the mean lag effect across the analyzed cities. All presented results are based on this number (four). The base mortality rate for each model is provided as the mortality rate for zero $TA_{Mag}$ (y-intercept of the regression model) indicating conditions of no thermal stress. This approach was also sensitivity-tested for seasonal variances in the mortality rate by the use of a seasonal de-trended, LOESS-smoothed (Cleveland, 1979) mortality time series instead of the crude mortality rate. Differences are negligible, which shows that the original approach chosen is insensitive to seasonal variances. In addition, HWE

occur usually during the summer month when mortality rates are low. For each regression model, the explained variance ($r^2$) was calculated. Error probabilities were calculated with a double-sided t-test. Regression models which were not statistically significant ($p > 0.05$) or comprised less than five HWE over the study period were discarded from further analyses. Error estimates for each regression model were calculated as the standard error of the regression coefficient ($RE_{RC}$) and of the base mortality rate ($RE_{BR}$). Regressions were also calculated for HWE with a minimum duration of consecutive days different from

three (1 to 5 days). The chosen minimum duration of three days yielded best results in terms of $r^2$, $RE_{BR}$, and $RE_{RC}$.

### 2.2.2 Multiple linear regressions

After detection of HWE, mean MDA8 ($MDA8_M$) were calculated for the total duration of each HWE. Multiple linear regressions (MLR) were then calculated using the ordinary least square error method with $TA_{Mag}$ and $MDA8_M$ of each HWE as predictor variables for mean mortality rates (as described in Sect. 2.2.1). The overall explained variance ($r^2$) and adjusted

explained variance ($r^2_{adj}$) as well as the explained variance for each single variable ($r^2_{TA_{Mag}}$ and $r^2_{MDA8_M}$) were calculated. An interaction term ($r^2_{TA_{Mag},MDA8_M}$) was also estimated as a cross-product effect of both predictor variables. Statistical significance is assumed for an error probability of $p < 0.05$, calculated with a double-sided t-test.

## 3 Results

Table 1 shows statistics for TA and MDA8 during the analysis period for each city. The 50th percentile of TA ranges from 10.0

°C in Hamburg to 11.9 °C in Cologne. For the analyzed cities, the highest recorded maximum TA is 31.1 °C in Cologne and the lowest 28.2 °C in Hamburg. The 50th percentile of the MDA8 concentration varies between 55.3 µg m$^{-3}$ in Frankfurt and 65.3 µg m$^{-3}$ in Leipzig. In two cities, Frankfurt and Cologne, by far the absolute highest MDA8 concentrations were recorded during the study period ($> 240$ µg m$^{-3}$, Table 1).

### 3.1 Regression analysis

Figure 2 presents results of the univariate regression analysis. For all cities, the analysis yields statistically significant results between $TA_{Mag}$ and mean mortality rates during HWE for a variety of $TA_{Thres}$. In all cities, statistically significant models are characterized by a minimum absolute $TA_{Thres}$ between 16 °C and 18 °C (Fig. 2, left panel). Results of all cities show generally increasing $r^2$ with increasing $TA_{Thres}$. Yet, differences across cities can be seen in the range of $TA_{Thres}$ and $r^2$ of the regression models. Highest values for $r^2$ are obtained for Berlin, Cologne, Frankfurt and Stuttgart with values of more than 60 % for

HWE of high $TA_{Thres}$. Cities with generally high TA (Table 1) also yield highest values of $r^2$. This may be a result of the





**Table 1.** Overview of city-specific statistics of the population (census corrected) at 31 December 2017. Statistics of air temperature and ozone concentrations are based on data from selected measurement sites during the years 2000 to 2017 (Table A1, Fig. A1). Cities are sorted from north to south, P refers to percentile. Sources: Population data: (DESTATIS, 2019), air temperature: (DWD, 2019), ozone concentrations: German Environment Agency (UBA), based on original data from air quality monitoring networks of the German federal states.

| City | Population | | | | Air temperature (daily average, °C) | | | Ozone (MDA8, µg m$^{-3}$) | | |
|---|---|---|---|---|---|---|---|---|---|---|
| | Total (No.) | under 18 (%) | over 65 (%) | Density (No. per km$^2$) | 50th P | 95th P | Max | 50th P | 95th P | Max |
| Hamburg | 1 800 865 | 17.7 | 18.7 | 2385 | 10.0 | 20.3 | 28.2 | 56.3 | 101.7 | 192.1 |
| Berlin | 3 542 728 | 17.5 | 19.6 | 3976 | 10.5 | 22.4 | 30.5 | 58.0 | 117.9 | 192.6 |
| Hanover | 529 957 | 15.8 | 19.0 | 2594 | 10.4 | 21.0 | 29.0 | 61.6 | 114.9 | 208.0 |
| Leipzig | 573 070 | 17.1 | 20.8 | 1924 | 10.3 | 21.8 | 29.0 | 65.3 | 123.5 | 198.3 |
| Cologne | 1 079 186 | 17.1 | 17.4 | 2665 | 11.9 | 22.6 | 31.1 | 56.8 | 121.4 | 240.1 |
| Frankfurt | 741 978 | 17.8 | 15.8 | 2988 | 11.6 | 23.2 | 30.7 | 55.3 | 122.7 | 240.5 |
| Stuttgart | 625 658 | 16.5 | 18.1 | 3018 | 11.1 | 22.7 | 30.3 | 62.3 | 129.1 | 203.7 |
| Munich | 1 451 696 | 16.6 | 17.8 | 4672 | 10.4 | 22.4 | 29.5 | 61.7 | 120.2 | 185.6 |

absence of HWE identified by higher $TA_{Thres}$ in cities like Hamburg, Hanover, Leipzig and Munich compared to the others. For all cities, increased mortality rates during HWE with $TA_{Thres} \leq 22$ °C can be explained by around 20 % of $TA_{Mag}$. Values of $RE_{BR}$ are low for each city ($< 0.2$) but show an increase towards higher $TA_{Thres}$. Values of $RE_{RC}$ are heterogeneous across different models as well as across different cities.

Whereas the range of absolute $TA_{Thres}$ of significant models varies across the cities, a percentile-based order reveals a more similar pattern in terms of threshold-$r^2$ relationship across the cities (Fig. 2, right panel). For HWE with $TA_{Thres} > 95$th percentile of the year-round TA distribution in 2000 to 2017, at least 20 % of the mortality rate can be explained by $TA_{Mag}$ across all cities. Except for Berlin and Cologne, $r^2$ is $< 20$ % for HWE with $TA_{Thres} < 95$th percentile. However, an increase in $r^2$ can be observed for all cities for HWE with $TA_{Thres} > 94$th percentile.

### 3.2   Multiple linear regression analysis

Results of the multiple linear regression and partitioning of $r^2$ are shown in Fig. 3. Generally, highest values are obtained for $r^2_{TA_{Mag}}$, increasing with increasing $TA_{Thres}$. This can be observed for almost all cities, while $r^2_{TA_{Mag}}$ values vary between cities. In particular, for HWE identified by higher $TA_{Thres}$ values variance in $TA_{Mag}$ alone explains at least 20 % up to 60 % of the mortality rates in Berlin, Cologne, Frankfurt and Stuttgart. Other cities show overall lower values of $r^2_{TA_{Mag}}$. Results also

reveal that in all cities the variance of mortality rates during these HWE can partly explained by the variance of $MDA8_M$, independently from $TA_{Mag}$. Particularly in Berlin, but also in Hanover and Stuttgart, mortality rates during HWE identified by high $TA_{Thres}$ cannot solely be explained by $MDA8_M$. However, this applies not to Frankfurt, where $r^2_{MDA8_M}$ values reach


**Figure 2.** Statistically significant results ($p < 0.05$, t test) from the univariate regression analysis with $TA_{Mag}$ as predictor variable for each city. Left panel x axis: absolute threshold value for HWE detection ($TA_{Thres}$), right panel x axis: percentile of $TA_{Thres}$ referring to the whole analysis period 2000 to 2017. y axis: explained variance of the models ($r^2$), relative errors of the base rate ($RE_{BR}$) and the regression coefficient ($RE_{RC}$), respectively.

higher values compared to $r^2_{TA_{Mag}}$, and in addition increase with increasing $TA_{Thres}$ (Fig. 3(f)). Differences between cities are also observable for the interaction term between both variables ($r^2_{TA_{Mag}, MDA8_M}$). Whereas some cities show only marginal values (Hamburg, Hanover, Munich), the others show an increasing interaction term with increasing $TA_{Thres}$, reaching up to 60 %






in Frankfurt. A different pattern for the interaction term is visible for Berlin. Highest values of $r^2_{\text{TA}_{\text{Mag}},\text{MDA8}_{\text{M}}}$ are obtained for medium $\text{TA}_{\text{Thres}}$ with declining trend towards higher $\text{TA}_{\text{Thres}}$.

## 4   Discussion

### 4.1   Relationship between $\text{TA}_{\text{Mag}}$ and mortality rates

The method used in this study allowed for an explorative identification and investigation of HWE, associated with an effect on mortality. In contrast to other investigations in the field of environmental epidemiology, the aim of this study was not to estimate air temperature or ozone related deaths. One of the main goals of this study was to identify HWE in multiple German cities, that are associated with increased mortality. In all cities, the strength of this association ($r^2$) increases with increasing $\text{TA}_{\text{Thres}}$. This is generally comparable with results from other investigations that show greater impact on mortality for more

intense HWE (e.g., Anderson and Bell, 2011; Tong et al., 2015).

However, the specific relationship between an absolute $\text{TA}_{\text{Thres}}$ and associated $r^2$ is affected by the specific TA distribution of each city and selected measurement site. Regression analyses were undertaken based on data of one selected measurement site per city, representing the atmospheric conditions of each city. Yet, it must be noted that data at these sites are not only influenced by city-wide characteristics, but also by characteristics of the closest environment at each site. Therefore, $\text{TA}_{\text{Thres}}$ is

affected by the distinct air temperature distribution of the selected measurement site and might differ for other locations. The usage of absolute $\text{TA}_{\text{Thres}}$ might thus be ambiguous for an inter-city comparison.

Throughout involved cities, an increase of $r^2$ was obtained around the 95th percentile of each city-specific TA distribution. This is also reported by the multi-city risk evaluation of various heat wave definitions for Australian cities (Tong et al., 2015). The use of relative $\text{TA}_{\text{Thres}}$ to identify HWE is thus suggested for studies investigating multiple cities to take into account

possible differences in TA distributions and acclimatization of the population to the local-specific air temperature distribution (Anderson and Bell, 2009, 2011; Tong et al., 2015). The use of the 95th percentile could thus be interpreted as one possibility to identify HWE that capture most of the mortality effect. It has to be stressed, though, that results also reveal statistically significant regression models for HWE identified with $\text{TA}_{\text{Thres}}$ lower than the 95th percentile. Such HWE, identified via $\text{TA}_{\text{Thres}}$ < 95th percentile, should thus likewise be considered as health relevant.

### 210  4.2   Relative contribution of $\text{TA}_{\text{Mag}}$ and $\text{MDA8}_{\text{M}}$ to mortality rates

Similar aspects as discussed above for the local dependence of air temperature measurements have to be noted also for ozone measurements. A comparison of regression analyses with the same method and based on data from different ozone measurement sites in Berlin was executed in (Krug et al., 2019). The ozone measurement site that was used in this study differs from the prior study. Yet, the data used here (Berlin Neukölln) revealed similar performance in terms of $r^2$ (Krug et al., 2019), but

is the closest to the co-located TA measurement site (Berlin-Tempelhof).


**Figure 3.** Results of the multiple linear regression (MLR) analysis between the predictor variables $TA_{Mag}$ and $MDA8_M$ and the mean mortality rate during HWE as independent variable. Each panel shows results for one city and different $TA_{Thres}$ (x axis). Top row of each panel shows overall $r^2$ (empty circles) and $r^2_{adj}$ (filled circles) of MLR models. Partitioned $r^2$ (y axis) are shown in the lower three rows for the predictors $TA_{Mag}$ (top, black), $MDA8_M$ (middle, light gray) as well as the interaction term of $TA_{Mag}$ and $MDA8_M$ (bottom, dark gray). Only results of statistically significant ($p < 0.05$) MLR models are displayed. Statistical significance of each predictor variable ($p < 0.05$) is marked with a star above each bar.





The second goal of this study was to investigate how ozone concentration contributes to mortality rates during HWE. MLR results between the predictors $TA_{Mag}$, $MDA8_M$ and mean mortality rates show that the latter is explained across all cities by up to 60 % by the variance of $TA_{Mag}$. This is in agreement with results of other studies which show that the effect of air temperature on mortality is stronger in comparison to the effect of ozone (e.g., Scortichini et al., 2018; Krug et al., 2019). $MDA8_M$ alone

partly explains mortality rates during HWE by up to 20 % in the investigated cities. Except of Frankfurt, this is mostly visible for HWE that are identified with low $TA_{Thres}$. Figure 4 shows that during HWE, MDA8 (per day) can reach values of up to 190 µg m$^{-3}$ (e.g. Fig. 4(e), Cologne). This exceeds the target value of 120 µg m$^{-3}$ set by the European Union to protect human health (EU, 2008). More than 50 % of the days during HWE identified via $TA_{Thres} < 20$ °C or even lower (depending on respective city) even fall below the ozone guideline value recommended by the World Health Organization (WHO) of 100

µg m$^{-3}$ (WHO, 2006). Associated adverse mortality effects during days with MDA8 values lower than the WHO guideline value for ozone were also found in the prior study focusing on Berlin (Krug et al., 2019) and in other studies and for other regions, e.g. Spain (Díaz et al., 2018) or cities in the United Kingdom (Atkinson et al., 2012; Powell et al., 2012).

However, the relative contribution of both $MDA8_M$ and $TA_{Mag}$ varies between cities and different $TA_{Thres}$. $MDA8_M$ explains more of the mortality rate at low $TA_{Thres}$ than $TA_{Mag}$. A lower $TA_{Thres}$ captures more HWE in which air temperature is relatively

low, but ozone concentrations can reach high values. This may occur during dry, sunny days in early summer, which promote the formation of ozone (Monks, 2000; Otero et al., 2016). This is also shown and discussed in (Krug et al., 2019). With increasing $TA_{Thres}$ a declining contribution of $MDA8_M$ alone to the mortality rate is observable (particularly in Berlin and Cologne, Fig. 3(b) and (e), respectively). An increasing contribution of the interaction term ($r^2_{TA_{Mag},MDA8_M}$) explaining mortality rates can be observed in all cities. This interaction is most pronounced in Berlin, Cologne, Frankfurt, Stuttgart and Leipzig and

indicates that ozone contributes to mortality rates during HWE identified by higher $TA_{Thres}$. Towards HWE identified by higher $TA_{Thres}$ air temperature becomes the more dominant factor explaining mortality rates and the variance of $MDA8_M$ is directly linked to the variance of $TA_{Mag}$. Similar conclusions were drawn by (Burkart et al., 2013) and is basically comparable to results that the mortality effect of ozone is strengthened during days of elevated air temperature and HWE (Vanos et al., 2015; Analitis et al., 2018; Scortichini et al., 2018).

## 240  4.3  Inter-city differences

Strongest associations between $TA_{Mag}$ as well as $MDA8_M$ and mortality rates were found for Berlin, Cologne, Frankfurt and Stuttgart. These cities are also those in which highest values of the 50th and 95th percentile and the maximum air temperature are recorded (Table 1). Based on absolute $TA_{Thres}$, it is not clear if the lower effect observed in Hamburg, Hanover, Leipzig and Munich are reasoned by the absence of HWE with $TA_{Thres} > 24$ °C (Leipzig, Munich) or $TA_{Thres} > 22$ °C (Hamburg,

Hanover), which occur in other cities and show strongest relationships to mortality rates.

Heterogeneities across cities were obtained not only for city-specific absolute $TA_{Thres}$ but also for their respective values of $r^2$, which is also reported in other studies investigating other cities across Europe (e.g., Filleul et al., 2006; Baccini et al., 2011; Burkart et al., 2013; Breitner et al., 2014; Analitis et al., 2018; Scortichini et al., 2018). City-specific peculiarities such as demographic or socio-economic characteristics at the community level may cause these differences (Stafoggia et al., 2006;

**Figure 4.** Top row of each panel: average number of HWE per year (y axis), detected per TA$_{Thres}$ (x axis). Bar-whisker plots display daily values of MDA8 during detected HWE. Boxes refer to the range between the 25th and 75th percentile (inter quartile range IQR), median values are given as solid lines, and whiskers are the minimum and maximum values excluding outliers (less than Q1-1.5·IQR, greater that Q3+1.5·IQR). Only results of statistically significant regression models are shown.




Anderson and Bell, 2011; Baccini et al., 2011). For instance, differences in age structure may influence the results. Elderly people were shown to be more vulnerable to heat (Yu et al., 2010; Scherer et al., 2013; Benmarhnia et al., 2015). Thus, a higher ratio of elderly people may strengthen the mortality rate during HWE. The ratio of the elderly over 65 years are in fact heterogeneous among involved cities (Table 1) but a linkage to city-specific relation to the effect on mortality rates cannot be deduced. Heterogeneities across cities may also be caused by local-specific geographical characteristics. The close distance to

the North and Baltic Sea, associated with a maritime climate, may prevent Hamburg from air temperatures that lead to higher impacts on mortality rates as observed for other cities. Similarly, Munich is not only the city situated at the highest altitude in this study but also closest to the Alps, which may influence the local weather conditions and lead to weather characteristics resulting in weaker relations between high air temperature and mortality rates. However, these reasons remain hypothetical and do not explain the low impacts in Hanover and Leipzig. To sum up, differences between cities are conceivable to be an overlay

of city-specific characteristics, such as demographic and geographic factors.

Results of this study underline the complexity to find similarities across different cities to determine appropriate criteria to identify hazardous episodes in terms of a health-related adverse effect, if there was such an effort. Some cities show a strong relationship between $TA_{Mag}$ and mortality rates, but these are also the cities experiencing highest air temperatures in this study (Table 1). Moreover, the strength of this relationship also varies across cities for equal $TA_{Thres}$ values. However, most

similarities arise by comparing results based on their local-specific percentile of the air temperature distribution rather than using absolute thresholds. This further also includes the interactive contribution of ozone.

Further research is needed to investigate local characteristics in more detail such as geographic drivers, socio-economic or socio-demographic factors which may affect the air-temperature-ozone-mortality relationship. These may cause local heterogeneities. Further, some studies also identified other air pollutants that affect mortality during HWE. Especially, concentrations

of particulate matter were also found to be increased during episodes of hot and dry weather (Tai et al., 2010; Schnell and Prather, 2017; Kalisa et al., 2018). Enhanced emission of secondary fine particles during hot weather conditions accompanied with reduced air movement may lead to this increased concentration especially in urban areas. Further, particulate matter is also associated with adverse mortality effect and is thus additionally relevant to human health during HWE (Burkart et al., 2013; Analitis et al., 2014; Schnell and Prather, 2017; Analitis et al., 2018).

**5 Conclusions**

This study investigated mortality rates during HWE in eight cities in Germany from 2000 to 2017. HWE were identified with a risk-based approach as a result of regressions between air temperature above a threshold and mean mortality rates during these episodes. HWE and thereby statistically significant regressions were detected in all selected cities for various air temperature thresholds. Results reveal a strong increase in the association around the 95th percentile of the local-specific air temperature

distribution. Apart from air temperature, ozone concentrations were shown to contribute to mortality rates during HWE. While air temperature was identified to be the dominant factor for elevated mortality rates, ozone concentrations alone contribute to those by up to 20 %. Additionally, results reveal that the effect of both stressors on mortality cannot be separated in many





cases, highlighting their strong interaction. Especially for HWE identified via higher threshold values of air temperature, ozone mostly contributes to mortality rates as statistically inseparable interaction with air temperature. To which extend air

temperature and ozone explain mortality rates differs across cities and for various air temperature thresholds. Some cities show weak associations while the contributions of both stressors to mortality rates are more pronounced in others.

This study underlines the complexity to deduce one universal threshold value in order to identify potentially hazardous HWE in terms of a health effect. Yet, it also emphasizes that besides air temperature ozone contributes to mortality during HWE in German cities. Future research should focus on city-specific characteristics such as population characteristics or geographical

peculiarities, which are likely leading to heterogeneities across cities and which may influence the respective air-temperature-ozone-mortality relationship.





**Appendix A**
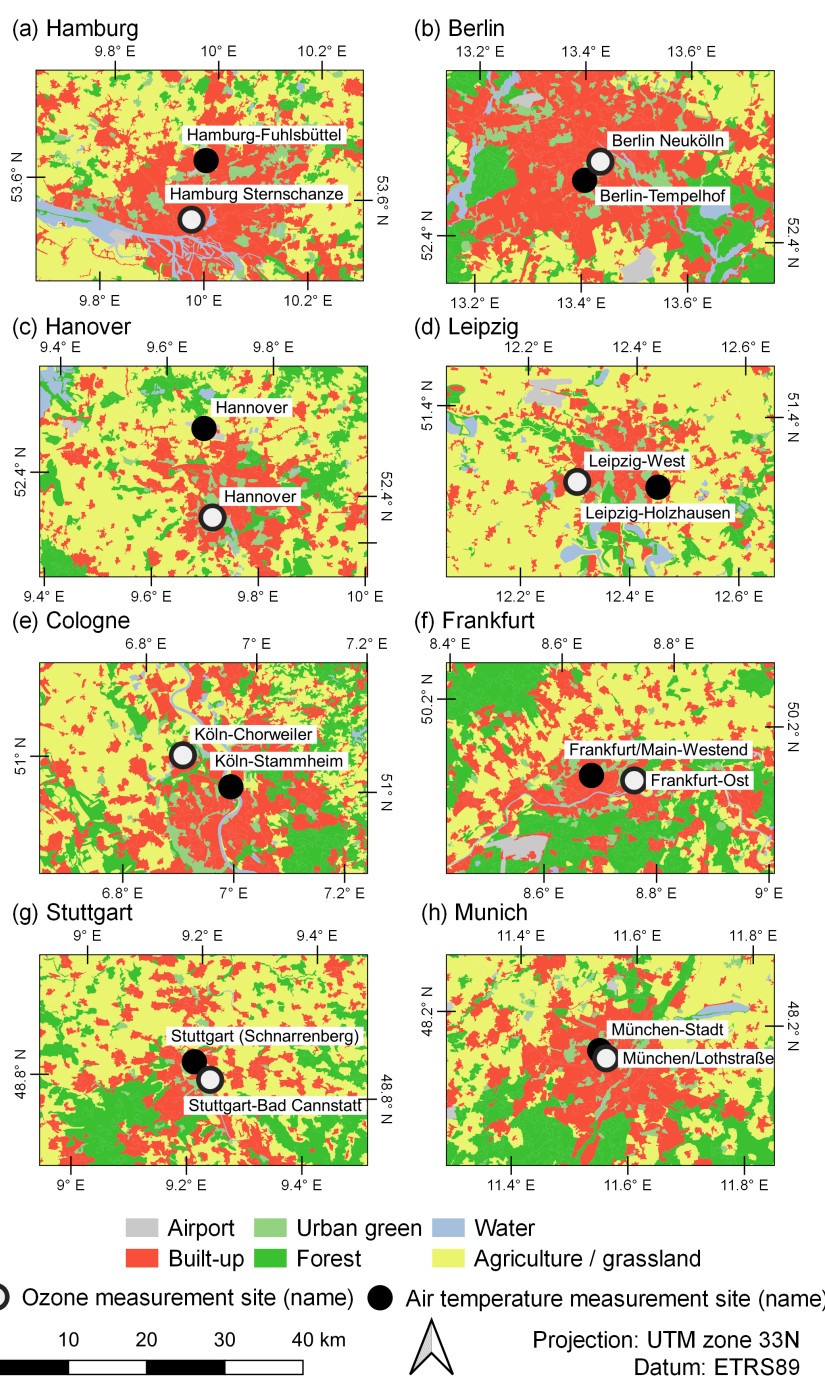

**Figure A1.** Location of selected air temperature and ozone measurement sites in the investigated cities. Land cover classification is based on CORINE 2018, v20 (EEA, 2019).




**Table A1.** Selected air temperature and ozone measurement sites of each city. Air temperature data were obtained from the German Weather Service (DWD) (DWD, 2019), data of ozone concentrations were obtained by the German Environment Agency (UBA), based on originally measured data of air quality networks of the German federal states.

| City | Air temperature measurements | | | | Ozone measurements | | | | Horizontal distance between co-located sites (km) |
| | Site name | Site location | | Elevation (m a.m.s.l.) | Site name | Site location | | Elevation (m a.m.s.l.) | |
| | | Longitude (° E) | Latitude (° N) | | | Longitude (° E) | Latitude (° N) | | |
|---|---|---|---|---|---|---|---|---|---|
| Hamburg | Hamburg-Fuhlsbüttel | 9.9881 | 53.6332 | 14 | Hamburg Sternschanze | 9.9679 | 53.5641 | 15 | 7.8 |
| Berlin | Berlin-Tempelhof | 13.4021 | 52.4675 | 48 | Berlin Neukölln | 13.4308 | 52.4895 | 35 | 3.1 |
| Hanover | Hannover | 9.6779 | 52.4644 | 58 | Hannover | 9.7061 | 52.3629 | 85 | 11.5 |
| Leipzig | Leipzig-Holzhausen | 12.4462 | 51.3151 | 138 | Leipzig-West | 12.2974 | 51.3179 | 115 | 10.4 |
| Cologne | Köln-Stammheim | 6.9777 | 50.9894 | 43 | Köln-Chorweiler | 6.8846 | 51.0193 | 46 | 7.3 |
| Frankfurt | Frankfurt/Main-Westend | 8.6694 | 50.1269 | 124 | Frankfurt-Ost | 8.7463 | 50.1253 | 100 | 5.5 |
| Stuttgart | Stuttgart (Schnarrenberg) | 9.2000 | 48.8281 | 314 | Stuttgart-Bad Cannstatt | 9.2297 | 48.8088 | 235 | 3.1 |
| Munich | München-Stadt | 11.5429 | 48.1631 | 515 | München/Lothstraße | 11.5547 | 48.1545 | 521 | 1.3 |



*Code availability.* Code can be made available by the authors upon request.

*Author contributions.* Alexander Krug conceived the concept and Dieter Scherer and Daniel Fenner gave technical and conceptual support.
Dieter Scherer provided the software. Alexander Krug collected the data, carried out the analyses, prepared the original draft of the manuscript and produced the visualizations. All authors gave support in the writing process, discussed the results, and commented on the manuscript. Dieter Scherer and Hans-Guido Mücke supervised the analysis.

*Competing interests.* The authors declare that they have no conflict of interest.

*Acknowledgements.* This research was funded by the Federal Ministry of Education and Research (BMBF), within the framework of Re-
search for Sustainable Development (FONA), as part of the consortium "Three-dimensional Observation and Modeling of Atmospheric Processes in Cities" (www.uc2-3do.org), under grant no. 01LP1912. This study was further supported by the doctoral research program of the German Environment Agency (UBA). Daniel Fenner received funding by the Deutsche Forschungsgemeinschaft (DFG) as part of the research project "Heat waves in Berlin, Germany - urban climate modifications" under grant no. SCHE 750/15-1. We kindly thank the section "Air Quality Assessment" of the German Environment Agency (UBA) for providing ozone data. We further express our gratitude to the
colleagues of the section "Environmental Medicine and Health Effects Assessment" for valuable discussions.



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
