# Peer review of "The contribution of air temperature and ozone to mortality rates during hot weather episodes in eight German cities during the years 2000 and 2017"

_Natural Hazards and Earth System Sciences, 2020_

## Referee Comment (RC1) · Anonymous Referee #1 · 2 Jun 2020

General comment

The manuscript examines the impacts of hot weather episodes (HWE) along with daily maximum eight-8hour moving average of surface ozone(MDA8) on rates mortality in eight cities in Germany. The study aims to compare the effects of HWE on the variability of mortality rates across the cities. In addition, MDA8 is included to examine the joined effect of both HWE and MDA8 on increase mortality rates. Based on a previous study, the authors used an event-based risk analysis. First, the HWE are iteratively detected using a sequence of thresholds. Then, regression analysis are applied to

evaluate the influence of the predictors (derived from the magnitude of the HWE and MDA8) on the variability of mortality rates. As expected, they found a significant contribution from HWE to mortality rates, as well as the strength of interaction HWE and MDA8. The results indicated that this effect was more pronounced in some specific cities (e.g. Berlin, Cologne). By using a simple methodology the results presented here are consistent with previous studies. Overall, the paper is well written and the methods and results are well presented and clear. The results are not totally surprising, but I think it provides important information regarding the definition of HWE on the basis of different thresholds and the impacts on mortality rates. In my opinion it will be of interest to the reader of NHESS. I have some minor comments that the authors should be able to address.

Specific comments

1. I have a comment regarding the period of the analysis. The study is performed to annual time series, and the authors tested long-term annual trends. But, given the strong seasonality of MDA8, which usually reaches the highest values in summer, I would expect the most important interaction HWE and MDA8 in summer. Did the authors take into consideration this?

2. Line 83: The analysis of HWE is based on daily average of air temperature (TA), and I understand that as in other studies, TA can be a suitable predictor. However, I was wondering if the authors have tested maximum temperature instead.

3. Line 233. In Berlin, it is observed a higher contribution from MDA8M at the lower TAThres, which is somehow surprising, since I would expect a higher contribution from MDA8 at higher TAThres. Why? The authors mention that it could due to stagnant conditions (dry, sunny days..) in early summer, but this is only observed in Berlin, do the authors have further explanations?

Technical corrections:

1. Line 128. It should be 0.5 °C.

2. Line 173. "Except for Berlin and Cologne, r2 is < 20 % for HWE with TAThres < 95th percentile", is that correct? I can see from figure 2 that r2 is larger for lower TAThres.

3. Line 180. "these HWE can partly explained", it should be "these HWE can be partly explained"

4. Line 228. "MDA8 explains more of the mortality rate at low TAt hres than TAMsg". I think it should be added where (e.g. Berlin), and refer to the figure to help the reader.

5. Line 230. As in my previous comment: "A lower TAThres captures more HWE..." where? All cities?

---

## Referee Comment (RC2) · Anonymous Referee #2 · 15 Jun 2020

The study examines the impact of hot weather episodes (HWE) on mortality counts for various German cities. Besides, the effect of ground-level ozone concentrations was taken into account. Multiple Linear Regressions with variable thresholds were used to establish the association between HWE and mortality counts. The study does not present a novel methodology (the approach has been published already in Scherer et al., and been applied by Fenner et al. and Jänicke et al.), nor does it reveal substantial new insights into the topic (the well-known relationships between heat and mortality). However, since the study gives temperature thresholds/ percentiles of strongest heatmortality connection for the cities under investigation it might be of value for some readers. In order to provide this information accurately, more details have to be given to the choice of the methods.

Specific comments: A major concern is related to the choice of Multiple Linear Regression (MLR) and the fact that all conclusions are based on the MLR test statistics assuming a normal distribution of the data. Crude mortality rates usually do not follow a normal distribution. If they do, please show results of normality testing. Mortality rates are count data and a Poisson distribution can be used as underlying distributional assumption in the scope of generalized linear models.

Furthermore, some of the conclusions have to be reconsidered. At page 11, lines 2018-2019 it is stated that the effect of air temperature on mortality is stronger in comparison to the effect of ozone. I disagree with this conclusion, because only events with high temperature have been selected and MLR is tuned towards this variable. These events do not necessarily go along with the ozone concentrations relevant for mortality. At lines 228ff. you notice that a lower TAThres captures more HWE in which air temperature is relatively low, but ozone concentrations can reach high values. This suggests that the typical non-linear relationship between temperature and ozone has an impact within your analysis. This should also be further investigated.

Technical corrections:

General comment: English should be revised by a native speaker.

Page1, line 21: replace "excessive mortality rates" by "excess mortality rates".

Page2, line 28: Please add NOx as further precursors.

Page2, line 50: replace "exposition" by "exposure". Page 5, line 127: remove "Firstly".

Section 2.2.2: add results of the F-test of the overall significance of the regression models.

Page 7, line 180: correct to "can partly be explained"
* * *

---

## Author Comment (AC1) · 9 Jul 2020

*Response to reviewer comment on "The contribution of air temperature and ozone to mortality rates during hot weather episodes in eight German cities during the years 2000 and 2017" by Alexander Krug et al.*

**Anonymous Referee #1**

We sincerely thank you for the overall feedback of our work as well as the constructive comments on the manuscript. This is highly appreciated. We reply to the reviewer comments below. Reviewer comments are in black and italic, authors' responses in blue.

*RC1: "I have a comment regarding the period of the analysis. The study is performed to annual time series, and the authors tested long-term annual trends. But, given the strong seasonality of MDA8, which usually reaches the highest values in summer, I would expect the most important interaction HWE and MDA8 in summer. Did the authors take into consideration this?"*

Answer: The method used in this study regresses between both process variables of air temperature magnitude ($TA_{Mag}$) and mean 8-hourly average ozone concentration ($MDA8_M$) as well as mean mortality rates as effect variable. Regressions are calculated only between the process and the effect during previously detected episodes of the whole analyzed period from 2000 to 2017. Although it was not aim of this study to investigate the effects of seasonal variances in air temperature or ozone concentrations, we fully agree with your expectation of increasing interaction during episodes of highest values of both air temperature and ozone. An indicator of possible higher impacts of interaction can be seen in Fig. 3 in the manuscript. At least four cities (Leipzig, Cologne, Frankfurt and Stuttgart) show increasing explained variance ($r^2$) of the interaction term with increasing air temperature threshold ($TA_{Thres}$). As episodes of exceeding high threshold values are likely to occur in mid-summer, results reveal the most interaction during these episodes (high $r^2$). Lower $TA_{Thres}$ includes an increasing number of episodes which are more likely to occur in early or late summer. A lower $r^2$ for the interaction term of these episodes can be seen in Fig. 3 of the manuscript. Concerning the other investigated cities, a lower $r^2$ of the interaction term for episodes of lower $TA_{Thes}$ is less visible, which is mainly due to lower values of the explained variances of these three variables. Yet, it cannot be excluded. We fully agree with your expectation of most pronounced interaction for episodes during the mid-summer time. Nevertheless, an investigation of different seasonal impacts or differences between distinct episodes are not in the focus of this study.

*RC2. "Line 83: The analysis of HWE is based on daily average of air temperature (TA), and I understand that as in other studies, TA can be a suitable predictor. However, I was wondering if the authors have tested maximum temperature instead."*

Answer: The decision to select daily average air temperature as predictor for mortality rates has been mainly made based on results of studies that are cited in section 2.1.1. According to your comment, we tested $T_{min}$ and $T_{max}$ as predictor variables in the regressions as well. The results for all investigated cities are displayed below. They reflect that $T_{max}$ is less suitable than $T_{min}$ and $T_{mean}$ to predict mortality rates during hot weather episodes (HWE), considering all cities. $T_{min}$ and $T_{mean}$ show higher values of $r^2$ than $T_{max}$. This confirms results of cited studies in section 2.1.1. and leads to our decision to use the daily average air temperature in our study. Furthermore, it reflects in a more general view, that night-time air temperature plays an important role for urban populations. $T_{mean}$ reflects the thermal situation of the entire day compared to $T_{min}$ or $T_{max}$. In addition to this, $T_{mean}$ better captures the urban heat island effect, which is commonly most pronounced in the first phase of the night and therefore not at the time of $T_{min}$ occurrence, commonly shortly before sunrise. According to these points, we selected $T_{mean}$ as thermal predictor for mortality in this study.

[Figure]

*Figure 1: Comparison of regression analysis based on different predictor variables $T_{min}$ (blue), $T_{mean}$ (black) and $T_{max}$ (red). Each panel displays results for one city. X axis: percentile of the respective air temperature distribution, y axis: explained variance ($r^2$) of regression models.*

*RC3. "Line 233. In Berlin, it is observed a higher contribution from MDA8M at the lower TAThres, which is somehow surprising, since I would expect a higher contribution from MDA8 at higher TAThres. Why? The authors mention that it could due to stagnant conditions (dry, sunny days..) in early summer, but this is only observed in Berlin, do the authors have further explanations?"*

Answer: In Fig. 3 of the manuscript, the light gray bars reflect the part of the variance of the mean mortality rate during episodes which can be explained by the variance of $MDA8_M$. Although all episodes were detected via air temperature thresholds, the regressions reveal that the variance of the mortality

rate cannot only be explained by the variance of the air temperature magnitude ($TA_{Mag}$), but mostly by the variance of $MDA8_M$. This is visible not only in results for Berlin, but also for Stuttgart and Cologne. The reason for this can be similarly discussed as we did in the answer to your comment RC1. Lower $TA_{Thres}$ capture more episodes, which occur in early or late summer. Especially in early summer, MDA8 can reach high values due to intense solar radiation and high photo-oxidative production rate. MDA8 values of up to 170 µg m³ for episodes of $TA_{Thres} \geq 16$ °C can be seen in Fig. 4. With increasing $TA_{Thres}$, the explained variance of $MDA8_M$ decreases and the air temperature becomes the most pronounced factor in explaining the variance of the mortality rate during these episodes. This does not mean, however, that MDA8 is not relevant for mortality rates during episodes of higher $TA_{Thes}$ in which the highest MDA8 concentrations may occur (Fig. 4). As shown in Fig. 3, the explained variance of $MDA8_M$ appears as a statistically inseparable part (the interaction term) of the variance of the air temperate magnitude ($TA_{Mag}$).

The specific contribution of $TA_{Mag}$, $MDA8_M$ and their interaction to mortality rates is spatially highly heterogeneous. Not only meteorological factors such as wind or humidity may influence the city specific relationship between these three variables. The topography or the emission rate of precursors through vegetation as well as population-specific factors (e.g. demography, socio-economy) may influence the city-specific relationship as well. This makes it more difficult to deduce similarities among different cities especially for the role of ozone. This fact mainly follows the results of other studies focusing on the relationship of air temperature, ozone and mortality, as cited and discussed in the manuscript. We will consider this comment in the discussion in sections 4.2. and 4.3. in the revision of the manuscript to make this clearer to the reader.

*Technical corrections:*

All your technical comments will be considered in the revised manuscript.

---

## Author Response (AR1)

*Response to comments by reviewer #1 and reviewer #2 on "The contribution of air temperature and ozone to mortality rates during hot weather episodes in eight German cities during the years 2000 and 2017" by Alexander Krug et al.*

We sincerely thank you for the overall feedback of our work as well as the constructive comments on the manuscript. This is highly appreciated. We reply to the reviewer comments below. Reviewer comments are in black and italic, authors' responses in blue. Please find the marked-up version of the manuscript at the end of this document.

**Anonymous Referee #1**

*Reviewer 1, Comment 1: "I have a comment regarding the period of the analysis. The study is performed to annual time series, and the authors tested long-term annual trends. But, given the strong seasonality of MDA8, which usually reaches the highest values in summer, I would expect the most important interaction HWE and MDA8 in summer. Did the authors take into consideration this?"*

Answer: The method used in this study regresses between both process variables of air temperature magnitude ($TA_{Mag}$) and mean 8-hourly average ozone concentration ($MDA8_M$) as well as mean mortality rates as effect variable. Regressions are calculated only between the process and the effect during previously detected episodes of the whole analyzed period from 2000 to 2017. Although it was not aim of this study to investigate the effects of seasonal variances in air temperature or ozone concentrations, we fully agree with your expectation of increasing interaction during episodes of highest values of both air temperature and ozone. An indicator of possible higher impacts of interaction can be seen in Fig. 3 in the manuscript. At least four cities (Leipzig, Cologne, Frankfurt and Stuttgart) show increasing explained variance ($r^2$) of the interaction term with increasing air temperature threshold ($TA_{Thres}$). As episodes of exceeding high threshold values are likely to occur in mid-summer, results reveal the most interaction during these episodes (high $r^2$). Lower $TA_{Thres}$ include an increasing number of episodes which are more likely to occur in early or late summer. A lower $r^2$ for the interaction term of these episodes can be seen in Fig. 3 of the manuscript. Concerning the other investigated cities, a lower $r^2$ of the interaction term for episodes of lower $TA_{Thes}$ is less visible, which is mainly due to lower values of the explained variances of these three variables. Yet, it cannot be excluded.

*Reviewer 1, Comment 2. "Line 83: The analysis of HWE is based on daily average of air temperature (TA), and I understand that as in other studies, TA can be a suitable predictor. However, I was wondering if the authors have tested maximum temperature instead."*

Answer: The decision to select daily average air temperature as predictor for mortality rates has been mainly made based on results of studies that are cited in section 2.1.1. According to your comment, we tested daily minimum air temperature (TN) and daily maximum air temperature (TX) as predictor variables in the regressions as well. The results for all investigated cities are displayed below and are also included in the appendix of the revised manuscript. They reflect that TX is less suitable than TN and TA to predict mortality rates during hot weather episodes (HWE), considering all cities. TN and TA show higher values of $r^2$ than TX. This confirms results of the cited studies in section 2.1.1. and leads to our decision to use TA in our study. Furthermore, it reflects in a more general view, that night-time air temperature plays an important role for urban populations. TA reflects the thermal situation of the entire day compared to TN or TX. In addition to this, TA better captures the urban heat island effect, which is commonly most pronounced in the first phase of the night and therefore not at the time of TN occurrence, commonly shortly before sunrise. According to these points, we selected TA as thermal predictor for mortality in this study. Following your suggestion, we enhanced the section 2.1.1 for this aspect.

[Figure]

*Figure 1: Comparison of regression analysis based on different predictor variables daily minimum air temperature (TN, blue), daily average air temperature (TA, black) and daily maximum air temperature (TX, red). Each panel displays results for one city. X axis: percentile of the respective air temperature distribution, y axis: explained variance (r²) of regression models.*

*Reviewer 1, Comment 3. "Line 233. In Berlin, it is observed a higher contribution from MDA8$_M$ at the lower TA$_{Thres}$, which is somehow surprising, since I would expect a higher contribution from MDA8 at higher TA$_{Thres}$. Why? The authors mention that it could due to stagnant conditions (dry, sunny days..) in early summer, but this is only observed in Berlin, do the authors have further explanations?"*

Answer: In Fig. 3 of the manuscript, the light gray bars reflect the part of the variance of the mean mortality rate during episodes which can be explained by the variance of MDA8$_M$. Although all episodes were detected via air temperature thresholds, the regressions reveal that the variance of the mortality rate cannot only be explained by the variance of the air temperature magnitude (TA$_{Mag}$), but mostly by the variance of MDA8$_M$. This is visible not only in results for Berlin, but also for Stuttgart and Cologne. The reason for this can be similarly discussed as we did in the answer to your comment RC1. Lower TA$_{Thres}$ capture more episodes, which occur in early or late summer. Especially in early summer, MDA8 can reach high values due to intense solar radiation and high photo-oxidative production rate. MDA8 values of up to 170 µg m³ for episodes of TA$_{Thres} \geq 16$ °C can be seen in Fig. 4. With increasing TA$_{Thres}$, the explained variance of MDA8$_M$ decreases and the air temperature becomes the most pronounced factor in explaining the variance of the mortality rate during these episodes. This does not mean, however, that MDA8 is not relevant for mortality rates during episodes of higher TA$_{Thes}$ in which the highest MDA8 concentrations may occur (Fig. 4). As shown in Fig. 3, the explained variance of MDA8$_M$ appears as a statistically inseparable part (the interaction term) of the variance of the air temperate magnitude (TA$_{Mag}$).

The specific contribution of TA$_{Mag}$, MDA8$_M$ and their interaction to mortality rates is spatially highly heterogeneous. Not only meteorological factors such as wind or humidity may influence the city specific relationship between these three variables. The topography or the emission rate of precursors through vegetation as well as population-specific factors (e.g. demography, socio-economy) may influence the city-specific relationship as well. This makes it more difficult to deduce similarities among different cities especially for the role of ozone. This fact mainly follows the results of other studies focusing on the relationship of air temperature, ozone and mortality, as cited and discussed in the manuscript. We considered this comment in the discussion in section 4.2 in the revised version of the manuscript to make this clearer to the reader.

*Technical corrections:*

1. *Line 128. It should be 0.5 °C.*

Changed.

2. Line 173. "Except for Berlin and Cologne, $r^2$ is < 20 % for HWE with $TA_{Thres}$ < $95^{th}$ percentile", is that correct? I can see from figure 2 that $r^2$ is larger for lower $TA_{Thres}$.

We removed this sentence.

3. Line 180. "these HWE can partly explained", it should be "these HWE can be partly explained"

We corrected this sentence.

4. Line 228. "MDA8 explains more of the mortality rate at low $TA_{Thres}$ than $TA_{Mag}$". I think it should be added where (e.g. Berlin), and refer to the figure to help the reader.

5. Line 230. As in my previous comment: "A lower TAThres captures more HWE. . ." where? All cities?

This section (4.2) was revised according to your and the second reviewer's specific and technical comments. Now, this section should be clearer.

**Anonymous Referee #2**

*Reviewer 2, Comment 1: "A major concern is related to the choice of Multiple Linear Regression (MLR) and the fact that all conclusions are based on the MLR test statistics assuming a normal distribution of the data. Crude mortality rates usually do not follow a normal distribution. If they do, please show results of normality testing. Mortality rates are count data and a Poisson distribution can be used as underlying distributional assumption in the scope of generalized linear models."*

Answer: The majority of epidemiological studies use generalized linear models to investigate the effect of air temperature or air quality on death counts. Crude death counts typically follow Poisson distributions which excludes the use of common linear regression analyzes. In contrast to this common approach, the underlying method of this study differs in two major points. Firstly, the method does not use crude death counts as effect variable. The mortality rate is used instead, which describes the number of deaths per population unit (mortality) as rate per time unit (day). Secondly, we do not investigate the overall relationship of daily air temperature values, ozone concentrations and death counts. Only episodes of variable duration of at least three days are investigated. For these episodes we assume a normal distribution for values of mortality rates of the investigated cites. Therefore, the method allows the use of simple or multiple regressions. A distribution histogram for the investigated cities as well as a table presenting results of the Shapiro-Wilk-Test is attached below.

We do not claim our method to be better than other approaches nor the best in terms to investigate air temperature or air quality effects on death counts or mortality rates, yet it allows a more precise identification of episodes of potentially hazardous atmospheric conditions for the public. Based on your comment we extended the section 2.1 for this aspect.

[Figure]

*Figure 2: Exemplary distribution of mean mortality rates during episodes exceeding 20 °C (daily average air temperature, TA) for at least three consecutive days. The sample contains episode-specific mean mortality rates of all cities. According to the Shapiro-Wilk-Test this distribution differs significantly from a normal distribution (p < 0.05).*

The table below provides results of the Shapiro-Wilk-Test for normality concerning mortality rates during respective hot weather episodes (HWE). The table contains all statistically significant models (t-test) of the univariate linear regression as used in the study (see sections 2.2.1, 3.1 and Fig. 2 in the manuscript). Results show inconclusive results of normal (50,4 % of all tested combinations) and significant non-normal (49,6 % of all tested combinations) tested distributions dependent on city and model-specific air temperature threshold. In conclusion, we keep the assumption of a near-normal distribution, acknowledging that our method is not the one ideal approach, or better than others, for investigations using mortality data. However, in terms of our research questions it delivers sufficient information about how to detect and characterize HWE as aimed in this study.

Table 1. Results of the Shapiro-Wilk-test for normal distribution of the mean mortality rate during hot weather episodes (HWE) exceeding $TA_{Thres}$. $p < 0.05$ (*) means not-normally distributed. City: city name, $TA_{Thres}$: Threshold temperature of the model, N episodes: number of episodes, W: W-value of the Shapiro-Wilk-test, $W_\alpha$: critical value.

| City | $TA_{Thres}$ | N episodes | W | $W_\alpha$ | p < 0.05 |
|---|---|---|---|---|---|
| Hamburg | 17.0 | 127 | 0.979 | 0.979 | * |
| Hamburg | 18.0 | 93 | 0.981 | 0.973 | |
| Hamburg | 19.0 | 59 | 0.978 | 0.960 | |
| Hamburg | 19.5 | 51 | 0.976 | 0.955 | |
| Hamburg | 20.0 | 47 | 0.969 | 0.952 | |
| Hamburg | 20.5 | 38 | 0.977 | 0.942 | |
| Hamburg | 21.0 | 35 | 0.975 | 0.938 | |
| Hamburg | 21.5 | 26 | 0.977 | 0.922 | |
| Berlin | 17.0 | 147 | 0.985 | 0.982 | |
| Berlin | 17.5 | 145 | 0.988 | 0.982 | |
| Berlin | 18.0 | 138 | 0.967 | 0.981 | * |
| Berlin | 18.5 | 133 | 0.965 | 0.980 | * |
| Berlin | 19.0 | 121 | 0.961 | 0.979 | * |
| Berlin | 19.5 | 116 | 0.954 | 0.978 | * |
| Berlin | 20.0 | 100 | 0.920 | 0.975 | * |
| Berlin | 20.5 | 86 | 0.903 | 0.971 | * |
| Berlin | 21.0 | 74 | 0.919 | 0.967 | * |
| Berlin | 21.5 | 67 | 0.928 | 0.964 | * |
| Berlin | 22.0 | 61 | 0.940 | 0.961 | * |
| Berlin | 22.5 | 47 | 0.906 | 0.952 | * |
| Berlin | 23.0 | 34 | 0.850 | 0.937 | * |
| Berlin | 23.5 | 28 | 0.865 | 0.926 | * |
| Berlin | 24.0 | 22 | 0.834 | 0.911 | * |
| Berlin | 24.5 | 16 | 0.872 | 0.887 | * |
| Berlin | 25.0 | 12 | 0.891 | 0.861 | |
| Hanover | 15.0 | 145 | 0.995 | 0.982 | |
| Hanover | 16.0 | 160 | 0.993 | 0.983 | |
| Hanover | 16.5 | 155 | 0.989 | 0.983 | |
| Hanover | 17.0 | 140 | 0.997 | 0.981 | |
| Hanover | 17.5 | 128 | 0.992 | 0.980 | |
| Hanover | 18.0 | 116 | 0.985 | 0.978 | |
| Hanover | 18.5 | 98 | 0.995 | 0.974 | |
| Hanover | 20.0 | 62 | 0.989 | 0.962 | |
| Hanover | 21.0 | 40 | 0.981 | 0.945 | |
| Hanover | 22.0 | 24 | 0.910 | 0.917 | * |
| Leipzig | 16.0 | 139 | 0.991 | 0.981 | |
| Leipzig | 16.5 | 153 | 0.989 | 0.983 | |

| | | | | | |
|---|---|---|---|---|---|
| Leipzig | 17.0 | 151 | 0.993 | 0.982 | |
| Leipzig | 17.5 | 148 | 0.994 | 0.982 | |
| Leipzig | 18.0 | 129 | 0.991 | 0.980 | |
| Leipzig | 18.5 | 118 | 0.988 | 0.978 | |
| Leipzig | 19.0 | 113 | 0.979 | 0.978 | |
| Leipzig | 19.5 | 101 | 0.973 | 0.975 | * |
| Leipzig | 20.5 | 66 | 0.976 | 0.964 | |
| Leipzig | 21.0 | 59 | 0.970 | 0.960 | |
| Leipzig | 21.5 | 47 | 0.944 | 0.952 | * |
| Leipzig | 22.0 | 38 | 0.949 | 0.942 | |
| Leipzig | 22.5 | 32 | 0.938 | 0.934 | |
| Leipzig | 23.0 | 21 | 0.927 | 0.908 | |
| Leipzig | 23.5 | 20 | 0.948 | 0.904 | |
| Leipzig | 24.0 | 15 | 0.924 | 0.882 | |
| Cologne | 16.5 | 149 | 0.973 | 0.982 | * |
| Cologne | 17.0 | 150 | 0.972 | 0.982 | * |
| Cologne | 17.5 | 149 | 0.975 | 0.982 | * |
| Cologne | 18.0 | 142 | 0.991 | 0.981 | |
| Cologne | 18.5 | 125 | 0.990 | 0.979 | |
| Cologne | 19.0 | 116 | 0.986 | 0.978 | |
| Cologne | 19.5 | 114 | 0.971 | 0.977 | * |
| Cologne | 20.0 | 98 | 0.912 | 0.974 | * |
| Cologne | 20.5 | 78 | 0.934 | 0.968 | * |
| Cologne | 21.0 | 67 | 0.879 | 0.964 | * |
| Cologne | 21.5 | 57 | 0.894 | 0.959 | * |
| Cologne | 22.0 | 50 | 0.915 | 0.954 | * |
| Cologne | 22.5 | 43 | 0.919 | 0.948 | * |
| Cologne | 23.0 | 34 | 0.863 | 0.937 | * |
| Cologne | 23.5 | 32 | 0.893 | 0.934 | * |
| Cologne | 24.0 | 22 | 0.839 | 0.911 | * |
| Cologne | 24.5 | 17 | 0.874 | 0.892 | * |
| Cologne | 25.0 | 11 | 0.832 | 0.855 | * |
| Cologne | 25.5 | 8 | 0.946 | 0.823 | |
| Cologne | 26.0 | 6 | 0.896 | 0.792 | |
| Frankfurt | 16.5 | 150 | 0.978 | 0.982 | * |
| Frankfurt | 17.0 | 146 | 0.974 | 0.982 | * |
| Frankfurt | 17.5 | 143 | 0.981 | 0.981 | * |
| Frankfurt | 18.0 | 146 | 0.964 | 0.982 | * |
| Frankfurt | 18.5 | 146 | 0.966 | 0.982 | * |
| Frankfurt | 19.0 | 130 | 0.969 | 0.980 | * |
| Frankfurt | 20.0 | 109 | 0.914 | 0.976 | * |
| Frankfurt | 20.5 | 100 | 0.840 | 0.975 | * |

| | | | | | |
|---|---|---|---|---|---|
| Frankfurt | 21.0 | 94 | 0.790 | 0.973 | * |
| Frankfurt | 21.5 | 80 | 0.830 | 0.969 | * |
| Frankfurt | 22.0 | 65 | 0.797 | 0.963 | * |
| Frankfurt | 22.5 | 54 | 0.808 | 0.957 | * |
| Frankfurt | 23.0 | 41 | 0.810 | 0.946 | * |
| Frankfurt | 23.5 | 39 | 0.776 | 0.944 | * |
| Frankfurt | 24.0 | 31 | 0.829 | 0.932 | * |
| Frankfurt | 24.5 | 25 | 0.781 | 0.920 | * |
| Frankfurt | 25.0 | 19 | 0.815 | 0.901 | * |
| Frankfurt | 25.5 | 14 | 0.859 | 0.875 | * |
| Frankfurt | 26.0 | 9 | 0.819 | 0.834 | * |
| Stuttgart | 18.0 | 131 | 0.989 | 0.980 | |
| Stuttgart | 18.5 | 116 | 0.982 | 0.978 | |
| Stuttgart | 19.0 | 111 | 0.990 | 0.977 | |
| Stuttgart | 19.5 | 101 | 0.961 | 0.975 | * |
| Stuttgart | 20.0 | 92 | 0.985 | 0.973 | |
| Stuttgart | 20.5 | 86 | 0.967 | 0.971 | * |
| Stuttgart | 21.0 | 76 | 0.960 | 0.968 | * |
| Stuttgart | 21.5 | 65 | 0.952 | 0.963 | * |
| Stuttgart | 22.0 | 58 | 0.962 | 0.959 | |
| Stuttgart | 22.5 | 48 | 0.955 | 0.952 | |
| Stuttgart | 23.0 | 36 | 0.952 | 0.940 | |
| Stuttgart | 23.5 | 31 | 0.904 | 0.932 | * |
| Stuttgart | 24.0 | 20 | 0.971 | 0.904 | |
| Stuttgart | 24.5 | 15 | 0.968 | 0.882 | |
| Stuttgart | 25.0 | 13 | 0.943 | 0.869 | |
| Stuttgart | 25.5 | 11 | 0.917 | 0.855 | |
| Stuttgart | 26.0 | 8 | 0.934 | 0.823 | |
| Munich | 16.0 | 164 | 0.987 | 0.984 | |
| Munich | 16.5 | 149 | 0.993 | 0.982 | |
| Munich | 17.0 | 143 | 0.976 | 0.981 | * |
| Munich | 17.5 | 132 | 0.984 | 0.980 | |
| Munich | 19.0 | 114 | 0.987 | 0.977 | |
| Munich | 19.5 | 107 | 0.995 | 0.976 | |
| Munich | 21.5 | 58 | 0.984 | 0.959 | |
| Munich | 22.0 | 51 | 0.969 | 0.955 | |
| Munich | 22.5 | 36 | 0.975 | 0.94 | |
| Munich | 23.0 | 26 | 0.971 | 0.922 | |

*Reviewer 2, Comment 2: "Furthermore, some of the conclusions have to be reconsidered. At page 11, lines 2018-2019 it is stated that the effect of air temperature on mortality is stronger in comparison to the effect of ozone. I disagree with this conclusion, because only events with high temperature have been selected and MLR is tuned towards this variable. These events do not necessarily go along with the ozone concentrations relevant for mortality."*

Answer: According to numerous investigations, ozone concentrations are highly relevant for public health and mortality. It is not the aim or conclusion of this study to weaken the importance of ozone concentrations. Our results underline quite the opposite; the found interaction underlines that during HWE not only elevated air temperature affects mortality rates. The interaction between air temperature and ozone concentrations as a statistically non-separable portion of the explained variance of mortality rates plays an important role during HWE. The statement you mentioned in your comment refers to the comparison of single proportions of each variable (air temperature magnitude, $TA_{Mag}$ and $MDA8_M$) to mortality rates. In particular, episodes detected via high $TA_{Thres}$, $TA_{Mag}$ explain more of the variance of the mortality rate that $MDA8_M$ (see Fig. 3 in the manuscript). We also investigated lower $TA_{Thres}$ down to the 70th percentile, in which $MDA8_M$ reaches higher values for the explained variance compared to $TA_{Mag}$. Nevertheless, we reconsidered the statements in section 4.2 according to your concerns.

*Reviewer 2, Comment 3: "At lines 228ff. you notice that a lower $TA_{Thres}$ captures more HWE in which air temperature is relatively low, but ozone concentrations can reach high values. This suggests that the typical non-linear relationship between temperature and ozone has an impact within your analysis. This should also be further investigated."*

Answer: Our results show that the relative contribution of $TA_{Mag}$, $MDA8_M$ and their interaction depends on the distinct $TA_{Thres}$ which identifies HWE (Fig. 3). In our opinion, different $r^2$ for $TA_{Mag}$ and $MDA8_M$ among different $TA_{Thres}$ indicate a potential non-linearity between air temperature and MDA8. Otherwise, as shown in Fig. 4 in the manuscript, the relationship between HWE and MDA8 shows a linearity for each $TA_{Thres}$, based on the position of the median, the 25th and the 75th percentile. Therefore, the interpretation of the role of air temperature and ozone concentrations strongly relates to the distinct $TA_{Thres}$ and thus the identification of potential hazardous episodes. Your comment is a truly interesting point for further investigations. However, in terms of our research questions and aims we see no need to extent our study to this point. Yet, your point will be taken into account in future work.

*Technical corrections:*

1. *General comment: English should be revised by a native speaker.*

We checked the manuscript and made several corrections.

2. *Page1, line 21: replace "excessive mortality rates" by "excess mortality rates".*

Replaced.

3. *Page2, line 28: Please add NOx as further precursors.*

Added.

4. *Page2, line 50: replace "exposition" by "exposure".*

Replaced.

5. *Page 5, line 127: remove "Firstly".*

Removed.

6. *Section 2.2.2: add results of the F-test of the overall significance of the regression models.*

All models of the multiple regression are now tested for overall statistical significance with a F-test. This led to minor changes in Fig. 3 and Fig 4. in the manuscript, but with no consequences for any results or conclusions. Furthermore, we adjusted sections 2.2.2 and 3.2 for this point.

7. *Page 7, line 180: correct to "can partly be explained"*

Corrected.

[revised manuscript text omitted]